# Willingness to Consume and Purchase Food with Edible Insects among Generation Z in Poland

**DOI:** 10.3390/foods13142202

**Published:** 2024-07-12

**Authors:** Anna Platta, Anna Mikulec, Monika Radzymińska, Stanisław Kowalski, Magdalena Skotnicka

**Affiliations:** 1Faculty of Management and Quality Science, Gdynia Maritime University, 81-87 Morska Street, 81-225 Gdynia, Poland; a.platta@wznj.umg.edu.pl; 2Faculty of Engineering Sciences, University of Applied Science in Nowy Sącz, 1a Zamenhofa Street, 33-300 Nowy Sącz, Poland; amikulec@ans-ns.edu.pl; 3Faculty of Economic Sciences, Institute of Management Science and Quality, University of Warmia and Mazury in Olsztyn, 4 Oczapowskiego St., 10-719 Olsztyn, Poland; mradz@uwm.edu.pl; 4Department of Carbohydrate Technology and Cereal Processing, Faculty of Food Technology, University of Agriculture in Krakow, 122 Balicka Street, 30-149 Krakow, Poland; rrkowals@cyf-kr.edu.pl; 5Department of Commodity Science, Faculty of Health Sciences, Medical University of Gdansk, 80-210 Gdansk, Poland

**Keywords:** entomophagy, insect-based foods, edible insects, novel food, willingness to consume, Generation Z

## Abstract

The consumption of insects (entomophagy) is attracting attention for economic, environmental and health reasons. The wide range of edible insect species, rich in protein, fat, minerals, vitamins and fibre, can play an important role in addressing global food insecurity. However, consumer acceptance remains a major barrier to the adoption of insects as a food source in many countries, including Europe. The aim of this study was to determine whether health and environmental concerns, attitudes and intentions towards purchasing edible insects and foods containing edible insects are associated with willingness to consume edible insects and foods containing edible insects among young consumers (Generation Z) in Poland. An empirical study was carried out in 2023, using a questionnaire with an indirect interview method via an online platform. On the basis of surveys conducted among Generation Z in Poland, it can be concluded that health and environmental concerns determine the willingness to consume selected products containing edible insects. At the same time, it should be noted that the more positive the respondents’ attitudes towards health and environmental concerns are, the greater their willingness to consume foods containing edible insects is. Attitudes and intentions towards purchasing foods containing edible insects were positively correlated with willingness to purchase and consume this type of food. The results obtained can contribute to efforts to promote the legitimacy of the production of new foods with edible insects in their composition.

## 1. Introduction

According to the predictions of the United Nations (UN), by the mid-century, the global human population is expected to exceed nine billion and will need about 70% more food to feed everyone [1]. Its production should take place without irreversible environmental destruction, and preferably with no or very limited expansion of the land already used for agriculture [2,3]. Rapid population growth and unsustainable dietary behaviour, coupled with pressure from climate change, threaten the food security of future generations and are critical issues that need to be addressed urgently [4].

With the deteriorating state of the environment, environmental concerns have captured the attention of the global public, and green consumption has become the strongest voice of the times. It is crucial that sustainable diets, defined by the Food and Agriculture Organization (FAO) as “nutritionally adequate, safe and healthy diets that meet the nutritional needs of present and future generations, respect biodiversity and ecosystems, ensure conservation, are culturally acceptable, accessible, affordable, nutritious, safe and healthy”, are adopted by members of societies of all generations [5,6,7]. As an alternative to food as a source of protein from traditional livestock and plant production, proteins derived from edible insects can be used. As part of an initiative to reduce world hunger and the negative environmental impact of animal husbandry, the European Union has created a specific action strategy in this area [8]. As early as 2013, the Food and Agriculture Organization of the United Nations began promoting insects as “an unexplored food source that can help address global food insecurity” [9]. As alternative protein sources come to the market, the most important factor in their success (or failure) will be what consumers think [10]. Consumers’ adoption of insects as a food ingredient could be a promising answer to the question of how we can feed a growing human population with a healthier and more sustainable source of protein [11,12,13,14,15,16].

The consumption of insects (entomophagy) has a long and rich history within human culture. The consumption of insects was largely dependent on cultural and regional factors, as well as the abundance and availability of specific species [17,18,19,20]. It is also noteworthy that entomophagy is not a novel practice in Europe. In ancient Rome, insects were consumed as a luxury food, but also as a response to food shortages. In many Western societies, including those influenced by Roman culture, the consumption of insects is not as common or culturally accepted as it once was [19,20,21]. It is worth to mention that already in 1885, Vincent M. Holt published a brief manifesto entitled *Why Not Eat Insects?* The author provided a list of insects suitable for consumption in Britain [22]. As in other global cultures, some insects in Europe were esteemed, others were disliked, and others were even feared [23,24]. For example, the Spanish fly, *Lytta vesicatoria* L., was commonly used in folk medicine in many parts of Europe [24,25,26,27]. The available literature indicates that there is a surprising wealth of historical evidence for the consumption of insects in Europe and in what is commonly referred to as Western culture. It has been documented that head lice were used as a medicine in Spain and among Hungarians in Romania [24,28,29]. There is some evidence that people consumed cockchafers (*Melolontha*), especially as food in times of famine. Furthermore, in some areas of Romania, Italy and Ireland, children probably also ate this type of insect [25,30]. It has been documented that salted or smoked grasshoppers were consumed in Russia and by Tartars in the Crimea until the 19th century [30]. In the 19th century, locusts were consumed as a food source in southern France [31]. The inhabitants of Wallachia and Moldavia consumed adult scarab beetles of the species *Amphimallon pini* (Ol.), while peasants in Lombardy utilised the beetle *Rhizotrogus assimilis* (Herbst) as a food source [32]. Additionally, there are still traditional dishes in European cuisine that contain insects, such as the Sardinian delicacy *casu marzu*. The cheese is the result of a decomposition process caused by the larvae of the cheese fly, *Piophila casei* (L.) [33]. The current increase in interest related to the use of insects as a sustainable and environmentally friendly source of protein and other nutrients represents an opportunity for the food industry and for ensuring global food security.

A number of studies on consumers’ acceptance of edible insects have been conducted in individual countries. One of these studies carried out on the Polish population was the study of Modlińska et al. [34]. As culture and social structure evolve, changes in individuals’ eating habits and environmental awareness occur. Generational differences strongly influence perspectives on sustainable and healthy eating [35]. Demographically, consumers can be divided into distinct groups by age, each of which exhibits homogeneous characteristics. Commonly recognised age classifications include the baby boomer generation, born between 1946 and 1964; Generation X, born between 1965 and 1979; Generation Y, also known as Millennials, born between 1980 and 1994; and Generation Z, born after 1995, often referred to as Generation C (‘connected’) or the post-millennial generation [36]. It has been observed that individuals belonging to different generations are characterised by different eating behaviours; for example, Generations X and Y prefer a so-called healthier approach to eating [37], while Generation Z is the most open and willing to shop based on hedonistic (personal, social and environmental) principles and values [37,38]. Generation Z’s interest in food and nutrition is correlated with physical body image [39] and the quality attributes of food [40]. Generation Z is believed to have been raised with a sense of mindfulness, which is reflected in their assertiveness and advocacy of their rights. In particular, Generation Z is characterised by increased concern about environmental change, escalating global terrorism and attacks, rising unemployment rates and income disparity [41]. From a marketing point of view, further research is needed to better understand the environmental concerns of Generation Z, as well as to expand current knowledge to develop specific products that are acceptable and meet their needs [42,43]. For it is Generation Z that accounts for 30% of future global purchasing and decision-making power, and it is young consumers who are the most influential trend-setting group in food production [44,45,46,47]. Numerous scientific studies have focused on identifying an individual’s motivations for specific actions, as motivations have been shown to directly and significantly influence current and future behavioural intentions [48]. According to the theory of planned behaviour (TPB) [49], an individual’s behaviour is shaped by his or her individual attitudes towards certain events or things. It has been claimed that four motives—value, career, learning and self-esteem—significantly influence the attitudes of Generation Z [50].

In order to increase people’s awareness and appreciation of the impact of diet on climate change and the environment, it is possible that a multidimensional approach, taking attitudes towards health, the environment, new foods containing edible insects and intentions to purchase such foods into account, would be beneficial. From this perspective, the relationship of sustainable eating behaviour involving edible insects and new food products containing edible insects with stated attitudes and intentions is a topic worth investigating. The aim of the study was to determine whether health and environmental concerns and attitudes and intentions towards purchasing edible insects and foods containing edible insects are associated with willingness to consume edible insects and foods containing edible insects among young consumers (Generation Z) in Poland.

The main research hypotheses were as follows:

**H1.** 
*Willingness to consume foods containing edible insects in the form of different product category groups can be linked to health concerns, environmental concerns and attitudes and intentions to purchase them.*


**H2.** 
*On the basis of attitudes towards health and environmental concerns and attitudes and intentions to purchase insect foods, readiness to consume foods containing edible insects in their composition can be predicted.*


## 2. Materials and Methods

### 2.1. Study Sample

The survey for which the results are presented in this article was conducted among students at five Polish higher education institutions using a survey questionnaire via an online platform. The empirical survey was conducted among 950 young consumers in the fourth quarter of 2023. In the surveyed group, 59.89% were women, 40.11% were men, 36.84% were rural residents, 17.68% were residents of towns with a population of up to 50,000, 16.53% were residents of towns with a population of 50,000 to 150,000 and 28.95% were residents of towns with a population of more than 150,000. All subjects gave their informed consent for inclusion before they participated in the study. The study was conducted in accordance with the Declaration of Helsinki, and the protocol was approved by the University Research Ethics Committee of the Cracow University of Economics (authorisation No. KEBN/71/0044/D24/2023). All respondents gave their free, informed consent to participate in the study and were assured of its anonymity. Participants in the study stated that they eat all foods, not limiting their consumption of meat or animal products.

### 2.2. Questionnaire and Data Analysis

The instrument contained 25 items, measured on a five-point Likert scale (1 = strongly disagree, 2 = disagree, 3 = no opinion, 4 = agree and 5 = strongly agree) [51].

The details of the items in each dimension were as follows: concern for health (3 items) [34]; concern for the environment (3 items) [34,52,53]; attitudes towards insect food (4 items) [54]; intention to purchase insect food (4 items) [55,56] and willingness to eat new foods containing edible insects (fresh, frozen, dried, powdered, e.g., meal) if they were available in the form of different products (11 items), namely (1) hamburgers, meat products and preparations [55,57,58]; (2) ready meals in the form of soups, pasta, pancakes, etc. [57,59,60,61]; (3) bread, rolls, pizza and other bakery products [57,62,63]; (4) cakes, cookies, chocolate-covered insects, and other pastry and confectionery products [34,57,62,64,65]; (5) bars, crisps, other snacks [57,66,67,68]; (6) post-workout drinks and nutritional supplements for people with high protein requirements [57]; (7) cottage cheese, yoghurt and other dairy drinks [57,69]; (8) sauces and mayonnaise [59]; (9) crickets (in frozen, dried or powdered form) [60,70]; (10) mealworm larvae (frozen, dried or powdered) [60,70] and (11) migratory locusts (frozen, dried or powdered) [70].

### 2.3. Statistical Methods

The questionnaire was validated by estimating the reliability of the scales used using Cronbach’s alpha coefficient; α was found to be in the range of 0.82–0.94, indicating the reliability of the scales used (Table 1).

The empirical material collected is presented in the form of the percentage distribution of the responses given, and selected descriptive statistics such as the median and standard deviation. The chi-square test with Yates’ correction was used to determine the relationships among environmental concerns, health concerns, attitudes towards foods containing edible insects, intentions to purchase such foods and willingness to consume selected products containing edible insects. Spearman’s rank correlation analysis was used to determine the relationships among health concerns, environmental concerns, attitudes and purchase intentions towards foods containing edible insects, and willingness to consume them.

A polynomial ordered logit model was constructed, in which the dependent variable was a variable examining willingness to consume products containing edible insects. The explanatory variable was calculated on the basis of variables related to willingness to consume edible insects (crickets in frozen, dried or powdered form; mealworm larvae in frozen, dried or powdered form; migratory locusts in frozen, dried or powdered form) and if they were available in the form of the following products: hamburgers, meat products and preparations; ready meals such as soups, pasta, pancakes, etc.; rolls, pizza, other bakery products; cakes, cookies, chocolate-covered insects, and other pastry and confectionery products; bars, crisps, other snacks; post-workout drinks and nutritional supplements for people with high protein requirements; cottage cheese, yoghurt and other dairy drinks; and sauces and mayonnaise. Three levels of readiness (low, ambivalent and high) to consume edible insects and products containing edible insects were distinguished according to the responses measured on a five-point Likert scale from 1 = strongly disagree to 5 = strongly agree [51]. The explanatory variables were health concerns, environmental concerns, purchase intention and attitudes. The objective variable was the ordinal variable of attitudes towards insect food (categories: negative, ambivalent and positive, as shown in Table 2). A validity table was used to check the quality of the model. The calculated R^2^ was approximately 71%.

A significance level of *p* < 0.05 was used for all statistical analyses. Calculations were performed using Statistica 13.3 (Tibco Software, Palo Alto, CA, USA).

## 3. Results

In the analysis of the variable “health concerns”, it was found that it significantly differentiated the students surveyed wo expressing willingness to consume “new foods” containing edible insects (fresh, frozen, dried, powdered, e.g., mealworms) if they were available in the form of ready meals such as soups, pasta, pancakes, sauces and others (*p* = 0.01); breads, rolls, pizzas or other bakery products (*p* = 0.01); post-workout drinks and nutritional supplements for people with high protein requirements (*p* < 0.01); and cottage cheese, yoghurt and other dairy drinks (*p* < 0.01). Persons with ambivalent attitudes towards health concerns significantly more often gave the answers “no” and “I don’t have an opinion” to the question regarding willingness to consume the abovementioned products compared with persons with negative and positive attitudes towards health concerns (Table 3). On the other hand, the answer “yes” for willingness to consume selected new edible insect products was significantly more frequently (at similar levels) expressed by people with ambivalent and positive attitudes towards health concerns (Table 3). Significant positive correlations (Spearman’s R) were also observed between health concerns and willingness to eat “new foods” containing edible insects (fresh, frozen, dried, powdered, e.g., mealworms) if they were available in the form of ready meals such as 1. soups, pasta, pancakes, sauces and others (0.07); breads, rolls, pizzas and other bakery products (0.07); cakes, cookies, chocolate-covered insects and other pastry and confectionery products (0.07); post-workout drinks and nutritional supplements for people with high protein requirements (0.10); and cottage cheeses, yoghurts and other dairy drinks (0.07) (Table 3). Thus, as concern for health increased, respondents expressed greater willingness to consume “new foods” containing edible insects. Furthermore, no significant relationship was found between students’ self-reported health concerns and willingness to consume edible insects such as crickets (in frozen, dried or powdered form), mealworm larvae (in frozen, dried or powdered form) and migratory locusts (in frozen, dried or powdered form) (Table 3).

Concern for the environment significantly differentiated respondents in expressing readiness to consume “new foods” containing edible insects (fresh, frozen, dried, powdered, e.g., mealworms) for all foods presented in the survey (*p* < 0.05). People with a low level of environmental concern were significantly less likely to express a willingness to consume foods with edible insects and significantly less likely to answer “I have no opinion” (Table 4). For all analysed products, a significant positive correlation was observed between environmental concern and willingness to consume “new foods” containing edible insects in their composition if they were available in the form of the food products proposed in the survey questionnaire. The Spearman’s R values ranged from 0.12 for hamburgers and processed meats to 0.18 for breads, rolls, pizzas and other baked goods; post-workout drinks and nutritional supplements for people with high protein requirements; and sauces and mayonnaise (Table 4). Thus, as concern for the environment increased, students were more likely to express a willingness to consume food containing edible insects in its composition. In addition, they expressed a willingness to eat edible insects such as crickets (in frozen, dried or powdered form; *p* < 0.01), mealworm larvae (in frozen, dried or powdered form; *p* = 0.01) and migratory locusts (in frozen, dried or powdered form; *p* < 0.01) (Table 4).

In the study group, a significant effect was observed between attitudes towards “new foods” containing edible insects and the expression of willingness to eat them. Those with positive attitudes were significantly more likely to express a willingness to consume insects, which was reflected in a significant moderately positive correlation between attitudes and willingness to consume this type of product. It ranged from 0.50 for mealworm larvae (in frozen, dried or powdered form) to 0.66 for ready meals such as soups, pasta, pancakes, sauces and others, as well as bread, rolls, pizza and other bakery products (Table 5). In addition, those declaring a positive attitude towards foods with edible insects were more than twice as likely to express a willingness to directly consume edible insects such as crickets, mealworm larvae and migratory locusts (*p* < 0.01) (Table 5).

A significant association was also observed in the study group of students between the intention to purchase and consume “new foods” containing edible insects and the expression of willingness to consume edible insects and foods containing them. Those demonstrating positive intentions were significantly more likely to express a willingness to consume foods with edible insects that were new to them. As observed in the case of attitudes towards “new foods” with edible insects, those displaying positive intentions were approximately twice as likely to express a willingness to consume edible insects directly (in addition to their use as an ingredient in the chosen product) (Table 6). Additionally, significant and moderate positive correlations were observed between intention and willingness to consume the edible insect foods presented in the study. The correlations determined ranged from 0.53 for mealworm larvae (in frozen, dried or powdered form) to 0.68 for ready meals (soups, pasta, pancakes, sauces and others) (Table 6). Consequently, individuals with positive intentions towards purchasing foods containing edible insects were more likely to indicate a willingness to consume not only foods containing edible insects but also the insects themselves (crickets, mealworm larvae and migratory locusts) (Table 6).

Table 7 presents a model showing the variables influencing willingness to eat edible insects and insect food. Assuming a significance level of 5%, the statistically significant variables for the model were found to be intentions to purchase insect foods (category: positive) and the attitudes presented towards these foods (categories: positive and ambivalent).

In order to interpret the individual variables, the odds ratio was used. This analysis allowed us to conclude the following.

Individuals with positive intentions towards purchasing food containing edible insects were over five times more likely to consume this type of food than those with negative intentions, all other things being equal.Individuals with positive attitudes towards food containing edible insects were over four times more likely to consume these types of foods compared with individuals with negative attitudes towards food containing edible insects, all other things being equal.Individuals with ambivalent attitudes towards food containing edible insects are approximately 30% more likely to consume such foods compared with individuals with negative attitudes towards food containing edible insects, all other things being equal.

## 4. Discussion

The analysis of the empirical research results obtained in this study enabled the researchers to achieve the study’s objective. The results of the quantitative studies conducted in the study allowed for the identification of factors significantly influencing the willingness of young consumers in Poland (Generation Z) to consume edible insects and food containing them. The empirical data presented in Table 7 and the Spearman’s rank correlations (Table 3, Table 4, Table 5 and Table 6) demonstrated that concern for the environment and attitudes and intentions to purchase edible insects and foods with insects play a significant role in explaining the willingness to consume edible insects and foods with insects observed in the study. The results of the study indicated that positive intentions and positive and ambivalent attitudes towards foods with edible insects in them were the main predictors of young consumers’ willingness to purchase and consume edible insects (crickets, migratory locusts and mealworm larvae) and foods containing them (hamburgers, meat products and preparations; ready meals such as soups, pasta, pancakes, sauces, etc.; bread, rolls, pizza and other bakery products; cakes, cookies, chocolate-covered insects, and other pastry and confectionery products; bars, crisps and other snacks; post-workout drinks and nutritional supplements for people with high protein requirements; cottage cheese, yoghurt and other dairy drinks; and sauces and mayonnaise).

The lack of data from different countries on young consumers’ attitudes and behaviour towards edible insects as food and foods containing edible insects in their composition did not allow a full examination of the results obtained. In light of the results obtained, it should be expected that those for whom environmental protection is important will display more positive attitudes towards insect food and a willingness to purchase it in the future. It is not insignificant that edible insects are a good source of nutrients in the human diet and that their husbandry also contributes to slowing down environmental degradation [71]. Consumers’ knowledge on this topic has been confirmed in other cross-sectional studies among adults over 40 years of age in Spain [72]. In order to introduce edible insects as a food in the future, it is important to inform adult consumers living in Europe, regardless of their age, about the health [58,72,73,74], environmental [58,72,74] and economic [58,72,75] benefits of including edible insects in their diet, as this may increase their willingness to include them in their diet. Other studies confirm that more and more consumers are becoming environmentally conscious [76] and treat sustainable consumption as their responsibility and behavioural intention [37,38]. Studies have emphasised that Generation Z, compared with other generations, is more consistent in its consumption choices [77] and is more aware of environmental issues [78] and sustainable eating [79]. In recent years, especially in Western European countries, where the concept of insect consumption is new, there has been a growing interest in the topic of the acceptance of entomophagy [34,40,80,81]. The growing insect-based food industry has been found to face significant challenges, particularly in overcoming consumers’ reluctance to eat insects [34,82,83,84,85,86,87,88]. The most studied and consistently identified factors explaining negative consumer attitudes towards entomophagy are disgust and food neophobia [34,62,89,90,91]. Our study and those of other authors in adult populations have identified the need to offer consumers products that make edible insects more familiar to Western society [72,92,93]. The production of commonly used flour-based products (bread, cakes, bars, etc.) [62,72,93] and offering culinary preparations closer to those of the regional culture [61,72] are ways to achieve this goal. In our research, we to identified specific products and product groups with edible insects that young Polish consumers (students) accepted and declared their willingness to consume. Consequently, the investigation of attitudes and intentions of young consumers to purchase and consume edible insect-based foods represents a pivotal aspect in the prospective evolution of entomophagy in Poland [40]. In particular, Generation Z is perceived to have greater environmental concerns and is willing to pay more for environmentally friendly products [94,95]. Furthermore, the literature has indicated that self-identity is a crucial factor in predicting the behavioural intentions of environmentally conscious consumers [96,97,98]. The concept of organic identity has been proposed as a key factor influencing consumers purchasing decisions in favour of organic products. This theory is supported by a range of empirical studies, including those by Sharma et al. [97], Whitmarsh and O’Neill [98], Khare and Pandey [96], and Mahasuweerachai and Suttikun [99]. Furthermore, the emergence of sustainability-oriented trends, such as the increased consumption of plant-based foods, the preference for organic produce, and the rise of the zero-waste movement, has been identified as a key factor influencing the dietary preferences of Generation Z [79]. The literature on Generation Z’s environmental behaviour patterns indicates that such a relationship exists [100]. The level of environmental information in society and the higher environmental awareness of the public are two key factors that influence the environmental behaviour of individuals [101,102]. Education [6,103] and finances [4,77,100] have an impact on the ecological consumer choices of Generation Z. Representatives of this group emphasise ecological aspects when making consumer choices [104,105,106]. Generation Z is less “me” oriented than their parents and grandparents [107]. They prefer sustainable fashion and, when grocery shopping, choose more environmentally friendly products [104,105,106,107,108,109,110,111,112], and they want to show their environmental behaviour on social media [113]. According to Atabek-Yiğit et al. [5], environmental literacy is the ability to understand and act to improve, restore and maintain the health of the environment. People with a high level of environmental literacy are important for preventing environmental problems and ensure sustainable development [114,115,116].

To induce consumers to accept insect-based foods, companies should focus on features that appeal to their target audience and explain how these foods benefit them. Further research is needed to understand Polish consumers’ attitudes towards the environment. Our study also showed that attitudes, intentions and environmental concern are important in predicting willingness to buy new foods.

The research has some limitations. One limitation is that the research focused exclusively on students representing the segment of young buyers in Poland. It is important to note that students may be more educated or more frequent travellers than their non-student peers. Such experiences may have influenced their openness and attitudes towards edible insects and foods containing them. On the other hand, it should be emphasised that this study was based on an extensive project investigating young consumers’ attitudes towards insect foods. The dispersion of consumers participating in the study was very large, as they came from every geographical region of Poland. As a result, the survey carried out in this research allowed for reliable validation of the scale used. The validated scale allowed a high degree of confidence in the data collected via the questions (exactly 25 items), both in this study and in future research. Therefore, it can be used to better understand the determinants affecting the acceptance of entomophagy by different population groups in Poland. It should also be emphasised that the willingness and readiness to purchase food containing edible insects in the survey questionnaire may differ from the actual results for consumption of food with edible insects.

As part of further research, it is planned to expand the research group to include representatives of Generation Z who are not currently involved in the educational system. Furthermore, it is planned to carry out a characterisation of the attributes of food with edible insects and to identify the factors determining and limiting the demand for edible insects and food with edible insects among Polish consumers. We believe that further research in this area is useful and justified in terms of predicting the importance of environmentally sustainable food production in the Polish population.

## 5. Conclusions

The objective of our study was to investigate the willingness of young Poles (students) to purchase and consume edible insects and food products containing insects. The research partially confirmed the research hypotheses (H1 and H2). The willingness to consume foods containing edible insects in the form of different groups of product categories was found to be related to environmental concerns, and attitudes and intentions to purchase them. On the other hand, concern for health was associated with the willingness to consume only certain products containing edible insects in their composition, such as ready meals (soups, pasta, pancakes, sauces, etc.); bread, rolls, pizza and other bakery products; cakes, pastries, chocolate-covered insects, and other pastry and confectionery products; post-workout drinks and nutritional supplements for people with high protein requirements; and cottage cheese, yoghurt and other dairy drinks.

The present study demonstrated that members of Generation Z in Poland are willing to purchase and consume edible insects. The present study makes a significant contribution towards the understanding of young consumers’ acceptance of entomophagy in Poland. This is because it provides precise information on the predictors that are important for Generation Z to consider when purchasing and consuming foods with edible insects. Furthermore, our survey identified a number of foods containing edible insects that young Polish consumers (students) accept and are willing to consume.

Our research has shown that environmental concerns and attitudes and intentions towards purchasing foods containing edible insects are important determinants of the potential willingness of Generation Z to purchase and consume both edible insects and foods containing them. Marketing communications for foods with edible insects should focus on communicating the environmental benefits of producing these foods. This is particularly important for building consumer awareness in this area. In addition, it will influence the formation of positive attitudes towards foods containing edible insects in different product categories, and thus the willingness to consume edible insects and foods with insects.

## Figures and Tables

**Table 1 foods-13-02202-t001:** Measurement model: reliability.

Construct	Cronbach α
Concern for health	0.82
Concern for the environment	0.86
Attitudes towards insect food	0.83
Intention to purchase insect food	0.94

**Table 2 foods-13-02202-t002:** Categories of changing attitudes towards insect food.

	Predicted		
Observed	Positive	Ambivalent	Negative
Positive	103	80	3
Ambivalent	47	341	76
Negative	1	72	227

**Table 3 foods-13-02202-t003:** Influence of the variable of health concerns on willingness to eat selected products containing edible insects.

Products			1 *	2	3	4	5	6	7	8	9	10	11
Attitudes of respondents towards health concerns	Negative	No	30.00	30.80	31.28	30.93	30.66	30.77	30.03	28.99	29.69	30.28	30.19
		No op. **	38.82	40.25	38.48	39.26	39.48	41.03	40.10	40.48	39.85	39.29	39.31
		Yes	31.18	28.95	30.25	29.81	29.86	28.21	29.87	30.53	30.46	30.43	30.49
	Ambivalent	No	32.23	28.36	27.54	25.95	25.74	29.66	25.93	29.08	24.63	26.67	26.62
		No op.	45.45	48.51	50.00	46.56	47.06	48.28	52.59	46.81	47.01	45.93	46.76
		Yes	22.31	23.13	22.46	27.48	27.21	22.07	21.48	24.11	28.39	27.41	26.62
	Positive	No	25.08	25.53	25.15	25.45	26.67	24.50	26.48	27.43	27.71	22.46	23.24
		No op.	40.44	36.78	38.65	39.07	38.41	34.90	32.88	35.40	36.14	39.13	38.03
		Yes	34.48	37.69	36.20	35.48	34.92	40.60	40.64	37.17	36.14	38.41	38.73
Statistics	Median	No	2.00	2.00	2.00	2.00	2.00	2.00	2.00	2.00	2.00	1.00	1.00
		No op.	2.00	2.00	3.00	2.00	2.00	2.00	2.00	2.00	2.00	2.00	2.00
		Yes	2.00	3.00	3.00	2.00	2.00	3.00	2.00	2.00	2.00	2.00	2.00
		R ***	0.05	**0.07**	**0.07**	**0.07**	0.05	**0.10**	**0.07**	0.04	0.05	0.04	0.05
		Chi^2^	7.53	13.12	12.63	6.31	5.74	20.72	19.58	7.94	4.99	6.94	7.51
		df	4	4	4	4	4	4	4	4	4	4	4
		*p*	*p* = 0.10	***p* = 0.01**	***p* = 0.01**	*p* = 0.18	*p* = 0.22	***p* < 0.01**	***p* < 0.01**	*p* = 0.09	*p* = 0.28	*p* = 0.14	*p* = 0.11

Explanatory notes: * 1, hamburgers, meat products and preparations; 2, prepared meals such as soups, pasta, pancakes, etc.; 3, bread, rolls, pizza and other bakery products; 4, cakes, cookies, chocolate-covered insects, and other pastry and confectionery products; 5, bars, crisps and other snacks; 6, post-workout drinks and nutritional supplements for people with high protein requirements; 7, cottage cheese, yoghurt and other dairy drinks; 8, sauces and mayonnaise; 9, crickets (in frozen, dried or powdered form); 10, mealworm larvae (frozen, dried or powdered); 11, migratory locusts (frozen, dried or powdered). ** I have no opinion. *** Spearman’s R; bold values are statistically significant

**Table 4 foods-13-02202-t004:** Influence of the variable of environmental concern on willingness to consume selected products containing edible insects.

Products			1 *	2	3	4	5	6	7	8	9	10	11
Attitudes of respondents towards environmental concerns	Negative	No	19.22	19.71	20.58	19.07	58.72	19.92	19.30	19.73	17.85	18.02	17.79
		No op. **	56.27	57.29	57.20	58.52	19.04	57.99	56.91	57.63	57.85	57.16	57.70
		Yes	24.51	23.00	22.22	22.41	22.24	20.09	23.99	22.64	24.30	24.82	24.51
	Ambivalent	No	9.09	8.96	9.42	9.92	8.83	11.03	12.59	10.64	8.21	9.63	9.35
		No op.	61.16	60.45	65.22	61.07	60.29	57.94	58.52	57.45	61.19	60.74	63.31
		Yes	29.75	30.60	25.36	29.01	30.88	31.03	28.89	31.91	30.60	29.63	27.34
	Positive	No	13.79	13.68	12.27	13.26	14.60	12.08	9.59	10.08	15.66	13.04	14.79
		No op.	53.92	52.58	50.61	49.11	50.16	52.01	52.97	51.33	45.18	46.38	41.55
		Yes	32.29	33.74	37.12	37.63	35.24	35.91	37.44	38.51	39.16	40.58	43.66
Statistics	Median	No	2.00	2.00	2.00	2.00	2.00	2.00	1.00	1.00	1.00	1.00	1.00
		No op.	2.00	2.00	2.00	2.00	2.00	2.00	2.00	2.00	2.00	2.00	2.00
		Yes	3.00	3.00	3.00	3.00	3.00	3.00	2.00	2.50	2.00	2.00	2.00
		R ***	**0.12**	**0.13**	**0.18**	**0.17**	**0.15**	**0.18**	**0.16**	**0.18**	**0.14**	**0.13**	**0.14**
		Chi^2^	12.63	19.88	32.50	26.39	23.58	24.83	21.95	29.39	22.44	19.07	27.02
		df	4	4	4	4	4	4	4	4	4	4	4
		*p*	***p* < 0.01**	***p* < 0.01**	***p* < 0.01**	***p* < 0.01**	***p* < 0.01**	***p* < 0.01**	***p* < 0.01**	***p* < 0.01**	***p* < 0.01**	***p* < 0.01**	***p* < 0.01**

Explanatory notes: * 1 to 11 are as in Table 3; ** I have no opinion; *** Spearman’s R; bold values are statistically significant.

**Table 5 foods-13-02202-t005:** Influence of the variable of attitudes towards insect food on willingness to eat selected products containing edible insects.

Products			1 *	2	3	4	5	6	7	8	9	10	11
Attitudes of respondents towards novel foods containing edible insects	Negative	No	40.59	42.92	43.00	39.26	41.48	40.43	35.23	36.36	32.15	31.31	31.69
		No op. **	50.78	51.33	50.41	51.85	52.10	50.10	52.52	52.66	54.00	54.36	54.56
		Yes	8.63	5.75	6.59	8.89	6.42	9.47	12.25	10.98	13.85	14.33	13.75
	Ambivalent	No	6.61	3.73	3.62	4.58	4.41	5.52	3.70	4.26	5.22	4.44	4.32
		No op.	77.69	79.10	80.43	74.81	75.74	76.55	72.59	70.92	70.90	67.41	67.63
		Yes	15.70	17.17	15.95	20.61	19.85	17.93	23.71	24.82	23.88	28.15	28.05
	Positive	No	2.20	2.43	2.45	1.43	2.86	3.02	3.20	1.77	3.61	2.90	2.82
		No op.	45.45	43.16	43.56	43.01	42.86	44.63	39.73	40.27	31.33	28.26	27.46
		Yes	52.35	54.41	53.99	55.56	54.28	52.35	57.07	57.96	65.06	68.84	69.72
Statistics	Median	No	1.00	1.00	1.00	1.00	1.00	1.00	1.00	1.00	1.00	1.00	1.00
		No op.	2.00	2.00	3.00	2.00	2.00	2.00	2.00	2.00	2.00	2.00	2.00
		Yes	4.00	4.00	4.00	4.00	4.00	4.00	4.00	4.00	3.00	3.00	3.00
		R ***	**0.62**	**0.66**	**0.66**	**0.63**	**0.64**	**0.60**	**0.57**	**0.59**	**0.53**	**0.50**	**0.52**
		Chi^2^	341.74	414.63	404.11	346.79	378.43	317.97	256.47	290.87	226.74	219.99	235.44
		df	4	4	4	4	4	4	4	4	4	4	4
		*p*	***p* < 0.01**	***p* < 0.01**	***p* < 0.01**	***p* < 0.01**	***p* < 0.01**	***p* < 0.01**	***p* < 0.01**	***p* < 0.01**	***p* < 0.01**	***p* < 0.01**	***p* < 0.01**

Explanatory notes: * 1 to 11 are as in Table 3; ** I have no opinion; *** Spearman’s R; bold values are statistically significant.

**Table 6 foods-13-02202-t006:** Effect of the variable of intention to purchase insect food on willingness to eat selected products containing edible insects.

Products			1 *	2	3	4	5	6	7	8	9	10	11
Respondents’ intentions to purchase novel foods containing edible insects	Negative	No	55.29	58.93	58.02	53.52	56.91	55.23	48.83	50.09	45.69	44.02	44.39
		No op. **	36.67	36.14	36.21	38.89	37.27	38.07	40.77	40.65	44.15	44.02	44.39
		Yes	8.04	4.93	5.77	7.59	5.82	6.70	10.40	9.26	10.16	11.96	11.22
	Ambivalent	No	23.97	20.90	21.01	21.37	20.59	21.38	20.00	19.15	20.15	19.26	18.71
		No op.	61.98	64.93	63.04	60.31	61.76	57.93	57.78	58.87	50.00	49.63	49.64
		Yes	14.05	14.17	15.95	18.32	17.65	20.69	22.22	21.98	29.85	31.11	31.65
	Positive	No	5.33	3.95	5.21	3.94	5.08	5.70	4.57	3.98	2.41	2.90	3.52
		No op.	46.08	44.38	44.79	43.01	44.13	44.30	40.18	39.38	33.13	31.88	30.28
		Yes	48.59	51.67	50.00	53.05	50.79	50.00	55.25	56.64	64.46	65.22	66.20
Statistics	Median	No	1.00	1.00	1.00	1.00	1.00	1.00	1.00	1.00	1.00	1.00	1.00
		No op.	3.00	3.00	3.00	2.00	3.00	3.00	2.00	2.00	2.00	2.00	2.00
		Yes	4.00	4.00	4.00	4.00	4.00	4.00	4.00	4.00	4.00	3.00	3.00
		R ***	**0.62**	**0.68**	**0.67**	**0.63**	**0.65**	**0.62**	**0.57**	**0.59**	**0.58**	**0.53**	**0.54**
		Chi^2^	335.28	434.87	387.09	345.46	379.38	334.10	261.53	295.94	266.12	218.37	232.16
		df	4	4	4	4	4	4	4	4	4	4	4
		*p*	***p* < 0.01**	***p* < 0.01**	***p* < 0.01**	***p* < 0.01**	***p* < 0.01**	***p* < 0.01**	***p* < 0.01**	***p* < 0.01**	***p* < 0.01**	***p* < 0.01**	***p* < 0.01**

Explanatory notes: * 1 to 11 are as in Table 3; ** I have no opinion; *** Spearman’s R; bold values are statistically significant.

**Table 7 foods-13-02202-t007:** Models indicating variables affecting willingness to eat edible insects and the eight product groups presented in the survey.

Variable	Category	Coef.	Std. Err.	Wald Test	95% Confidence		*p*-Value	OR
Free expression 1		−2.399	0.134	322.655	−2.661	−2.137	**0.000**	0.091
Free expression 2		1.520	0.129	139.854	1.268	1.772	**0.000**	4.573
Health concern scale	Positive	−0.022	0.111	0.038	−0.239	0.195	0.845	0.979
Health concern scale	Ambivalent	−0.031	0.101	0.094	−0.228	0.166	0.759	0.970
Environmental care scale	Positive	0.155	0.126	1.532	−0.091	0.402	0.216	1.168
Environmental care scale	Ambivalent	−0.104	0.103	1.031	−0.305	0.097	0.310	0.901
Intentions	Positive	1.682	0.153	121.529	1.383	1.981	**0.000**	5.378
Intentions	Ambivalent	0.053	0.103	0.267	−0.149	0.255	0.605	1.055
Attitudes	positive	1.467	0.160	84.553	1.154	1.780	**0.000**	4.336
Attitudes	ambivalent	0.259	0.108	5.782	0.048	0.470	**0.016**	1.296

Explanatory notes: bold values are statistically significant.

## Data Availability

The raw data supporting the conclusions of this article will be made available by the authors on request.

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
