# Peer review of "Willingness to Consume and Purchase Food with Edible Insects among Generation Z in Poland"

_foods, 2024, doi:10.3390/foods13142202_

Round 1
Reviewer 1 Report
Comments and Suggestions for Authors
The aim of the study « Willingness to consume and purchase food with edible insects among Generation Z in Poland» was to determine whether health and environmental concerns and attitudes and intentions towards purchasing edible insects and foods containing edible insects are associated with willingness to consume edible insects and foods containing edible insects among young consumers (Generation Z) in Poland.
Furthermore, in 2021, colleagues from Poland published a similar article titled "Relationship between Acceptance of Insects as an Alternative to Meat and Willingness to Consume Insect-Based Food—A Study on a Representative Sample of the Polish Population." What unique contributions does this manuscript offer that distinguish it from the aforementioned article?
Integrating a bit of history on entomophagy in the West into the Introduction would also be compelling. The removal of insects from diets in Western countries historically relates to their seasonal unavailability and also carries religious implications.
The results showed in the tables are challenging to comprehend. Firstly, would it be beneficial to transpose each table for better clarity? The formatting is quite complex. Additionally, in the tables, particularly in the second and third lines of the titles (e.g., "no opinion"), it is noted that in the third line, N stands for negative attitude, A for ambivalent attitude, and P for positive attitude. Could you clarify how the line appears when respondents have no opinion? I would appreciate clarification across all tables in the "Results" section where these scenarios are present. I would like to review the reorganized tables before finalizing my review.
Author Response
We would like to thank Editor and all Reviewers for their efforts in the evaluation of the manuscript and for valuable comments. Please find enclosed answers to questions and suggestions of reviewers. We hope that after introduced changes, manuscript will be suitable for publication.
The aim of the study « Willingness to consume and purchase food with edible insects among Generation Z in Poland» was to determine whether health and environmental concerns and attitudes and intentions towards purchasing edible insects and foods containing edible insects are associated with willingness to consume edible insects and foods containing edible insects among young consumers (Generation Z) in Poland.
Furthermore, in 2021, colleagues from Poland published a similar article titled "Relationship between Acceptance of Insects as an Alternative to Meat and Willingness to Consume Insect-Based Food—A Study on a Representative Sample of the Polish Population." What unique contributions does this manuscript offer that distinguish it from the aforementioned article?
Answer: We believe that both works, although they refer to similar issues, actually focus on the analysis of different information obtained in the research. In our opinion, these works neither exclude nor duplicate each other, but constitute an interesting complement for the reader. Study on a representative sample of the Polish population in 2021 (Modlinska, K.; Adamczyk, D.; Maison, D.; Goncikowska, K.; Pisula, W. Relationship between acceptance of insects as an alternative to meat and willingness to consume insect-based food - A study on a representative sample of the Polish population. Foods. 2021, 10, 2420. https://doi.org/10.3390/foods10102420) showed that acceptance of insects as an alternative to meat (general perspective) does not transfer into willingness to buy and consume them (individual perspective). The authors analysed the population cross-sectionally, referring to a wide spectrum of consumers. Furthermore, the authors found that consumers in Poland who declare their acceptance of insects as a meat substitute may not be willing to purchase insects for consumption. Our study concerted on investigating the willingness of young Poles (students) to purchase and consume edible insects and food with insects. In our study, we proved that representatives of generation Z in Poland show a willingness to purchase and consume edible insects (crickets, migratory locusts, mealworm larvae) and foods with insects (hamburgers and processed meats, ready meals, bread and other bakery products, pastries and confectionery, sweets and salty snacks, post-workout drinks and nutritional supplements for people with high protein requirements, dairy products and sauces and mayonnaise). Our study is an important contribution to understanding the acceptance of entomogaphy by young consumers in Poland. This is because it provides precise information on what predictors are important for generation Z to buy and consume foods with edible insects dedicated to them. In this context, it can be said that our work is a more detailed analysis of a specific group of consumers consisting of young people undertaking university studies. This means that this work neither replicates nor questions the achievements presented in the work of Modlińska et al. However, it is a certain extension and supplement to it.
Integrating a bit of history on entomophagy in the West into the Introduction would also be compelling. The removal of insects from diets in Western countries historically relates to their seasonal unavailability and also carries religious implications.
Answer: Thank you for your valuable comment. The information in the introduction has been completed.
The results showed in the tables are challenging to comprehend. Firstly, would it be beneficial to transpose each table for better clarity? The formatting is quite complex.
Answer: Thank you very much for your valuable comment. We have made changes, we hope that now the tables are more readable.
Additionally, in the tables, particularly in the second and third lines of the titles (e.g., "no opinion"), it is noted that in the third line, N stands for negative attitude, A for ambivalent attitude, and P for positive attitude. Could you clarify how the line appears when respondents have no opinion? I would appreciate clarification across all tables in the "Results" section where these scenarios are present. I would like to review the reorganized tables before finalizing my review.
Answer: We apologies for the mistake, there was a mix-up of lines when labelling attitudes and responses. This has been corrected in the tables.
Reviewer 2 Report
Comments and Suggestions for Authors
The manuscript entitled "Willingness to consume and purchase food with edible insects among Generation Z in Poland" provides valuable insights into the attitudes of young Polish consumers towards edible insects. However, I have some specific observations that I believe would enhance the robustness and impact of the manuscript:
1. While this work is not presented as a novel methodological proposal or groundbreaking study, the information it provides is highly valuable. Given that similar methodologies have been applied and replicated across various regions, especially in Europe and the Western world, it is essential to compare these results with those obtained by other research groups. Specifically, comparing attitudes of different generational groups towards entomophagy across various studies would help to either support or refute some of the conclusive observations of this study. This comparative analysis would offer a more robust perspective on young consumers' attitudes towards insect-based foods.
2. Furthermore, the limitations mentioned by the authors should include a perspective on the broader representativeness of Generation Z, particularly those not engaged in educational systems. This demographic is often underrepresented in such studies but may exhibit different attitudes influenced more by socio-cultural factors, prejudices, and ideologies than by formal education. Notably, populations with varying educational levels can have contrasting viewpoints, often shaped by their socio-cultural environment rather than the rationality and logic fostered by formal education. Addressing this aspect would provide a more comprehensive understanding of the target demographic.
By incorporating these comparative and broader representational considerations, the manuscript would significantly strengthen its contributions to the field and provide a deeper, more nuanced understanding of young consumers' willingness to adopt entomophagy.
Thank you for considering my review.
Author Response
We would like to thank Editor and all Reviewers for their efforts in the evaluation of the manuscript and for valuable comments. Please find enclosed answers to questions and suggestions of reviewers. We hope that after introduced changes, manuscript will be suitable for publication.
The manuscript entitled "Willingness to consume and purchase food with edible insects among Generation Z in Poland" provides valuable insights into the attitudes of young Polish consumers towards edible insects. However, I have some specific observations that I believe would enhance the robustness and impact of the manuscript:
- While this work is not presented as a novel methodological proposal or groundbreaking study, the information it provides is highly valuable. Given that similar methodologies have been applied and replicated across various regions, especially in Europe and the Western world, it is essential to compare these results with those obtained by other research groups. Specifically, comparing attitudes of different generational groups towards entomophagy across various studies would help to either support or refute some of the conclusive observations of this study. This comparative analysis would offer a more robust perspective on young consumers' attitudes towards insect-based foods.
Answer: Thank you for your valuable comment. The discussion of the results has been supplemented with references to results obtained by other research groups. Conducting an analysis of generational groups is a highly advisable and interesting idea. We selected a special group of consumers for our research, i.e. students. We assumed that this is a group of people with greater knowledge and social awareness, as well as more open to new things than their peers who do not undertake studies. Of course, also in other generations, e.g. in the generation of older people with different life attitudes, the results obtained would be different. It would certainly be interesting to compare different generational groups in this aspect, which may become the subject of our next research.
- Furthermore, the limitations mentioned by the authors should include a perspective on the broader representativeness of Generation Z, particularly those not engaged in educational systems. This demographic is often underrepresented in such studies but may exhibit different attitudes influenced more by socio-cultural factors, prejudices, and ideologies than by formal education. Notably, populations with varying educational levels can have contrasting viewpoints, often shaped by their socio-cultural environment rather than the rationality and logic fostered by formal education. Addressing this aspect would provide a more comprehensive understanding of the target demographic. By incorporating these comparative and broader representational considerations, the manuscript would significantly strengthen its contributions to the field and provide a deeper, more nuanced understanding of young consumers' willingness to adopt entomophagy.
Answer: Thank you for your valuable comment. We are aware of this limitation, which we have taken into account when describing the limitations of the study In future research we also plan to include representatives of Generation Z not involved in the education system. The information used has been included in the text.